# Intrahepatic Cholestasis of Pregnancy during COVID-19 Pandemic

**DOI:** 10.3390/medicina60040676

**Published:** 2024-04-22

**Authors:** Paulina Malarkiewicz, Urszula Nowacka, Aleksandra Januszaniec, Alicja Mankiewicz, Szymon Kozłowski, Tadeusz Issat

**Affiliations:** 1Department of Obstetrics and Gynecology, School of Medicine, Collegium Medicum of the University of Warmia and Mazury, al. Warszawska 30, 10-082 Olsztyn, Poland; 2Department of Obstetrics and Gynecology, Institute of Mother and Child, Kasprzaka 17a, 01-211 Warsaw, Poland; 3Department of Obstetrics and Gynecology, Olsztyn Specialist Hospital, Żołnierska 18; 10-561 Olsztyn, Poland

**Keywords:** intrahepatic cholestasis of pregnancy, pregnancy, COVID-19, liver damage, maternal–fetal medicine, pandemic

## Abstract

*Background and Objectives*: Intrahepatic cholestasis of pregnancy (ICP) stands as one of the most prevalent concerns in maternal–fetal medicine, presenting a significant risk to fetal health and often associated with liver dysfunction. Concurrently, the coronavirus-19 (COVID-19) infection can lead to hepatic cell injury through both direct and indirect pathways. Hypothetically, these two conditions may coincide, influencing each other. This study aimed to comparatively assess the incidence and severity of ICP before and during the COVID-19 pandemic. *Methods*: A retrospective cohort study was conducted, comparing the incidence and severity of ICP between January 2018 and February 2020 (pre-COVID-19 period) and March 2020 to March 2022 (COVID-19 period) across two hospitals, encompassing 7799 deliveries. The diagnosis of ICP was established using the ICD-10 code and defined as total bile acids (BA) levels ≥ 10 μmol/L. Statistical analysis included descriptive statistics, Chi-square and Mann–Whitney U tests, as well as multiple or logistic regression analysis. *Results*: A total of 226 cases of ICP were identified. The incidence of mild cholestasis (BA < 40 μmol/L) was lower during the pandemic compared to before (3% before versus 2%, *p* < 0.05), while the incidence of moderate and severe ICP remained unchanged (0.6% before vs. 0.4%, *p* = 0.2). Overall, the total incidence of ICP was lower during the pandemic (3.6% before versus 2.4%, *p* = 0.01). No significant differences were observed in severity (as defined by BA and liver function test levels), rates of caesarean section, or neonatal birth weights. *Conclusions*: During the COVID-19 pandemic, the total incidence of ICP appeared to be lower. However, this reduction was primarily observed in cases of mild ICP, potentially indicating challenges in detection or reduced access to medical services during this period. The incidence of moderate and severe ICP remained unchanged, suggesting that these forms of the condition were unaffected by the pandemic’s circumstances.

## 1. Introduction

Intrahepatic cholestasis of pregnancy (ICP) is the most common reversible hepatic disorder related to pregnancy, characterized by pruritus in the absence of a primary skin condition, with onset predominantly in the third trimester [1]. Although the condition poses a mild risk to a woman, it carries a major risk for a fetus, including preterm delivery, meconium-stained amniotic fluid, and stillbirth [2]. The prevalence differs among ethnic groups and geographical regions, varying between 0.3 and 5.6% of pregnancies. Incidence seems to be more pronounced in winter months [3,4]. Around 25% of women may experience itching in pregnancy; therefore, diagnosis cannot be made based on symptoms alone [5]. To diagnose ICP, maternal bile acid levels must exceed a given threshold, recently established by the Royal College of Obstetricians and Gynaecologists to be 19 μmol/L, while non-fasting [1]. The commonly used upper limit of bile acids of 10 μmol/L currently lies within the normal range, and concomitant itching should be diagnosed as gestational pruritus [6]. Although elevated liver enzymes may be increased simultaneously, they do not reflect the risk of fetal demise as bile acids do [1,7]. According to a meta-analysis with aggregate and individual patient data, the risk of stillbirth in singleton pregnancies is associated with the maximum total bile acid concentration, especially over 100 μmol/L [8]. Ursodeoxycholic acid, a first-line treatment until now, does not reduce adverse perinatal outcomes and may even increase the readings of the bile acids [9]. ICP is associated with pre-existing metabolic derangements, including maternal dyslipidemia and glucose intolerance; therefore, it is often diagnosed with gestational diabetes mellitus and pre-eclampsia [4].

Although ICP typically resolves almost immediately after delivery, there is growing evidence that the diagnosis increases the risk of maternal hepatobiliary disease later in life, in addition to susceptibility to hepatobiliary cancer, immune, and cardiovascular diseases. Therefore, women who have had a pregnancy complicated by ICP should undergo close follow-up [4]. Apart from the fetal adverse outcomes, in utero exposure to high bile acids, exacerbated by the reversal of the trans-placental gradient of bile acid concentrations, may play a role in fetal programming and metabolic changes later in life, which can already be observed in adolescence [4]. All these factors emphasize the importance of a definitive diagnosis of ICP during pregnancy.

The coronavirus (COVID-19, COVID) pandemic outbreak significantly transformed medical management, affecting not only general medical conditions but also psychological aspects [10]. Due to multiple immune system alterations during gestation, pregnant individuals become susceptible to pathogens, especially respiratory viruses, making them a high-risk group during the pandemic [11]. Maternal infection during pregnancy increases the risk of adverse outcomes, particularly preterm birth [12]. Although vaccination against COVID-19 has an established role in preventing maternal and fetal complications, the rate of hesitancy among pregnant women remains substantial [11]. Among the myriad effects on various targets, SARS-CoV2 has been found to cause direct and indirect hepatic injury, especially in those with chronic illnesses and immunocompromised states [13]. COVID-19 disease is associated with hepatic involvement not only during the initial infection but also as sequelae [14,15,16]. Regarding pregnancy, elevated liver enzymes have been reported in patients with COVID infection and intrahepatic cholestasis of pregnancy (ICP) compared to non-infected women [17]. As some authors have hypothesized a possible correlation–causation sequence between maternal COVID-19 and ICP [18], the objective of this study was to assess the incidence and course of ICP before and during the pandemic era.

## 2. Materials and Methods

This was a retrospective, two-site cohort study conducted at tertiary centers for maternal–fetal medicine and obstetrics, specifically the Institute of Mother and Child (Instytut Matki i Dziecka, IMiD) in Warsaw, Poland, and Olsztyn Specialist Hospital (WSS) in Olsztyn, Poland, among individuals receiving prenatal and/or perinatal care at general obstetrics, midwifery, and maternal–fetal medicine clinics. We included pregnant women with maximum total bile acids levels ≥ 10 μmol/L (according to the national guideline at that time) without underlying liver disease, in singleton and twin pregnancies [19]. Multiples with more than two fetuses, individuals with underlying chronic liver diseases, pre-eclampsia, hemolysis, elevated liver enzymes and low platelet counts (HELLP), viral hepatitis, and acute fatty liver of pregnancy were excluded from this study. Eligible individuals were enrolled from March 2020 (the onset of the pandemic in Poland) to March 2022. The pre-COVID-19 period was defined as January 2018 to February 2020. Data details were obtained from hospital data systems, and the following aspects were collected: parity, COVID infection during pregnancy (prior to the diagnosis of ICP), confirmed by a positive result of real-time polymerase chain reaction (RT-PCR) from nasopharyngeal and/or oropharyngeal samples derived from a national registry, maximum levels of ALT (alanine transaminase), ASP (aspartate transaminase), and BA (bile acids); pharmacological agents (ursodeoxycholic acid and others), gestational age at diagnosis, number of fetuses, concomitant diseases, labor (spontaneous, induced, no labor), delivery (vaginal, elective cesarean, emergency cesarean), gestational age at delivery, and birth weight. This study was approved by the Bioethics Committee at the Faculty of Medicine, University of Warmia and Mazury (approval number: 13/2022). The analysis was conducted in two variants. Variant one (1) compared the incidence of ICP and characteristics of the ICP patients of the pre-pandemic period with the pandemic period. The start date of the COVID-19 period in Poland was established as 4 March 2020 (the date of the first confirmed case in Poland). In variant two (2), the analysis compared ICP patients with a confirmed diagnosis of COVID-19 with the remaining number of ICP patients, regardless of the period. Statistical analysis was performed using Statistica 13.0, employing descriptive statistics, statistical tests, and multiple regression analysis (for endpoints with a quantitative variable) or logistic regression analysis (for endpoints with a binary variable). Categorized variables were compared using the χ^2^ test, and quantitative variables were compared using the Mann–Whitney test. The level of significance was established as 0.05.

## 3. Results

The analysis included 226 ethnically homogeneous women with ICP. Among these, 122 (54%) delivered during the pre-pandemic period and 104 (46%) during the pandemic period. A positive COVID-19 test result was obtained by 25 women (11% of the total number and 24% of the pandemic period). The basic characteristics of the study group are shown in Table 1. Patients with ICP in both analyzed variants did not differ in maternal age, primiparity, time of ICP diagnosis, and rate of concomitant gestational diabetes mellitus (GDM). The comparative characteristics of the study population are shown in Table 2. There were 3397 deliveries before and 4402 during the pandemic era in both centers. The incidence of ICP seemed to be lower during the pandemic (3.6% versus 2.4%, *p* = 0.01). Although the incidence of mild ICP (BA < 40 μmol/L) appeared to be lower during the pandemic (3% before vs. 2%, *p* < 0.05), the incidence of moderate and severe ICP (BA ≥ 40 μmol/L) remained stable (0.6% before vs. 0.4% during the pandemic, *p* = 0.2).

No differences were observed between levels of BA, AST, and ALT between the groups, both on admission and at maximum reported levels (Table 2). ICP was the main reason for the induction of labor (IOL) and cesarean section in a similar proportion of patients with ICP before and during the pandemic (52% vs. 53%, *p* = 0.778), as well as in those with COVID infection during pregnancy and those without (53% vs. 50%, *p* = 0.782). The rate of instrumental deliveries (cesarean sections, vacuum-assisted deliveries, or forceps deliveries) also remained similar (47% before and 48% during the pandemic, *p* = 0.905; 53% without COVID and 50% with COVID infection; *p* = 0.782). There were no differences observed in birth weight between both variants (3250 g before and 3180 g during the pandemic, *p* = 0.263; 3135 g without COVID and 3200 g with COVID infection; *p* = 0.925).

## 4. Discussion

This study yielded several findings as follows: (1) the incidence of total ICP appeared to be lower during the pandemic; (2) the incidence of mild ICP seemed to be lower during the pandemic; (3) the incidence of moderate and severe ICP remained stable; (4) the timing of ICP diagnosis did not differ between the pre-pandemic and pandemic periods, nor between COVID-infected and non-infected patients; (5) the levels of BA, ALT, and AST remained similar in patients with ICP during both the pre-pandemic and pandemic periods, and between COVID-infected and non-infected patients; (6) ICP was the reason for induction of labor/cesarean section in a similar proportion of patients in both scenarios; (7) the rate of instrumental deliveries remained similar during both the pre-pandemic and pandemic periods, and between COVID-infected and non-infected patients; and (8) neonatal birth weight did not differ between the groups in both scenarios.

The healthcare crisis induced by COVID-19 has underscored the necessity for an interdisciplinary approach to identify potential sequelae and risk groups for complications. Multiple studies have reported altered liver function tests (LFTs) not only in the general population infected with COVID-19, but also in pregnant individuals [20,21,22]. Approximately 3–5% of pregnancies can result in abnormal LFTs in the second half of pregnancy due to various reasons besides ICP, such as pre-eclampsia, hemolysis, elevated liver enzymes and low platelet counts, viral hepatitis, or acute fatty liver of pregnancy [23]. These conditions were excluded from our study.

Our data revealed a lower incidence of ICP during the pandemic compared to the pre-pandemic period. Due to the scarcity of available studies, drawing a definitive conclusion is challenging; however, this observation specifically pertains to patients with mild ICP. We hypothesize that women with only a mild increase in BA levels and subtle symptoms may not have sought medical advice due to the COVID-19 pandemic. According to the Centers for Disease Control and Prevention (CDC), 40% of adults were avoiding medical care due to concerns related to COVID infection [24]. While there is one study showing contradictory results, comparing both conclusions is challenging due to different ethnic settings and timing [25]. In the latest study on the topic by Holden et al., the authors reported an increased incidence of COVID infection in women with ICP. Among 596 patients, all individuals who tested positive for COVID-19 with ICP were Hispanic [26].

The severity of ICP, as reflected by liver function test (LFT) results, did not differ between the pre-pandemic and pandemic periods, nor between COVID-infected and non-infected patients. Partially similar results were reported by Soffer et al. [27]. In their study, BA levels remained stable, and AST and ALT levels were higher during the pandemic, regardless of ongoing detected COVID infection. However, it is extremely difficult to compare the infection rate within the pregnant population of the studies. Although it has been established that COVID infection could lead to hepatic involvement and subsequent injury, the similar BA results suggest a different pathophysiological mechanism. Januszewski et al. [28] examined the liver damage profile in COVID-infected pregnant patients of an identical ethnic background to our study population. In that group, the median level of bile acid in COVID-infected patients was higher compared to healthy pregnant individuals, but levels were within normal ranges even in women with the most severe course of COVID-19 disease. In those with liver injury, the type of damage was rather hepatotoxic than cholestatic. The liver damage in COVID-19 patients is multifactorial, as the spectrum of liver injury in COVID infection ranges from direct infection with SARS-CoV-2 to indirect involvement by systemic inflammation, hypoxic changes, iatrogenic causes such as medications, intensive care procedures, and exacerbation of underlying liver disease [8]. Cai et al. [20] suggested that liver injury and abnormal tests are mostly due to certain medications used during hospitalization; therefore, it may apply mostly to patients with severe COVID-19. Both sites of this study did not accept pregnant patients with severe COVID-19, thus there was a possible sampling bias. Moreover, age is a known risk factor for a severe COVID-19 course, and in general, the pregnant population consists of healthy individuals in reproductive age [29]. Another factor is that different medications are avoided by physicians in the pregnant population, or, although prescribed, they are avoided by the women themselves [30].

There are several limitations to this retrospective study in addition to those mentioned above. Many patients could have had asymptomatic COVID infection or may have intentionally avoided testing. We lack details of the total number of COVID infections in all pregnancies due to differing universal testing policies between the study sites. We did not include information about vaccination as the majority of data were collected before universal vaccination recommendations for pregnant women [11]. Moreover, we did not stratify the severity of COVID infection. The time span between COVID infection and ICP was not reported, as infection at any point during pregnancy was taken into account. We did not separate a subgroup of ongoing infection and ICP.

Nevertheless, there are several strengths to mention. We included only patients who fulfilled laboratory criteria for ICP. The presence of COVID infection was derived from the national registry, where only confirmed laboratory results from swabs taken by healthcare professionals were entered. All patients fulfilling the ICP criteria were tested for COVID-19 upon admission. To our knowledge, this is the first full-text study comparing the characteristics of ICP before and during the pandemic in our geographical and racial setting. However, it is evident that the existing literature lacks large-scale studies examining the incidence of cholestasis of pregnancy, particularly in diverse settings. To address this gap, comprehensive analyses of individual patient data from various settings are imperative.

## 5. Conclusions

During the COVID-19 pandemic, the total incidence of ICP appeared to be lower. However, this decrease was primarily observed in cases of mild ICP, suggesting potential challenges in detection or avoidance of medical services during the pandemic, as these patients may have had less intense symptoms.

## Figures and Tables

**Table 1 medicina-60-00676-t001:** Baseline characteristics of the study population.

		Variant 1			Variant 2		
	General	Childbirth before the Pandemic	Childbirth during the Pandemic	*p*-Value for the Difference	No History of COVID-19	History of COVID-19	*p*-Value for the Difference
Age (years): median IQR	32.4 (29.0–36.4)	32.4 (29.6–36.3)	32.6 (28.9–36.4)	0.979 *	32.5 (29.1–36.4)	31.3 (28.9–36.5)	0.571 *
Primiparity	101 (51%)	59 (52%)	42 (51%)	0.873 **	88 (51%)	13 (52%)	0.938 **
Diagnosis of ICP—days before the due date	34 (21–50)	34 (22–46)	33 (21–55)	0.801 *	33 (21–48.5)	40 (22–58.0)	0.118 *
Twins, N (%)	16 (7%)	9 (7%)	7 (7%)	0.970 **	1 (3%)	15 (7%)	0.412 **
Gestational diabetes, N (%)	65 (29%)	32 (25%)	33 (32%)	0.209 **	57(28%)	8(27%)	0.872

* Mann–Whitney U test; ** Pearson’s chi-square test.

**Table 2 medicina-60-00676-t002:** Main outcomes of this study.

	Before Pandemic ^&^	% ^$^	In the Pandemic ^&^	% ^$^	*p* Value for the Difference	Without COVID-19 ^$^	% ^$^	COVID-19 + ^$^	% ^$^	*p* Value for the Difference
Number of deliveries	3397		4402		0.0013 **					
Incidence of ICP	122	3.59% (359/10,000)	104	2.36% (236/10,000)		201		25		
Moderate ICP ≥ 40 μmol/L	19	0.559%	16	0.36%	0.1913 **	31		4		
Mild ICP < 40 μmol/L	103	3.03%	88	2.00%	0.0035 **	170		21		
BA (μmol/L) -on admission -maximum	13.9 18.5	(9.0–30.0) (10.9–40.9)	16.9 22.5	(10.2–31.5) (12.2–44.5)	0.301 * 0.178 *	14.6 20.2	(9.4–31.1) (13.3–42.3)	15.1 19.5	(9.6–28.0) 12.6–43.0)	0.972 * 0.600 *
ALT (U/l) -on admission -maximum	132.0 158.0	(57.0–122.0) (76.0–301.0)	132.5 154.5	(65.0–269.0) (86.0–359.0)	0.833 * 0.522 *	139.0 157.0	(65.0–276.0) (84.0–325.0)	101.0 149.5	(51.0–215.0) (66.0–288.0)	0.171 * 0.528 *
AST (U/l) -on admission -maximum	83.5 99.0	(41.0–149.5) (50.5–173.5)	78.5 95.0	(47.0–151.0) (51.0–198.0)	0.793 * 0.631 *	84.5 100.0	(41.0–153.5) (57.0–178.0)	63.5 84.5	(47.0–94.0) (49.0–176.0)	0.435 * 0.666 *
ICP as a main indication for IOL or CS	67	51.5%	55	53.4%	0.778 **	107	(52.7%)	15	(50.0%)	0.782 **
Cesarean section/vacuum assisted/forceps delivery	58	47%	47	48%	0.905 **	88	(45%)	17	59%	0.199 **
Birth weight (grams)	3250	(2800–3600)	3180	(2720–3540)	0.263 *	3135	(2650–3750)	3200	(2770–3540)	0.925 *

ICP—intrahepatic cholestasis of pregnancy; BA—bile acids; AST—aspartate aminotransferase; ALT—alanine aminotransferase; IOL—induction of labor; CS—cesarean section; ^&^ number for categorical variables, median for continuous variables; ^$^ percent for categorical variables, interquartile range for continuous variables; * Mann–Whitney U test; ** Pearson’s chi-square test.

## Data Availability

The datasets used and/or analyzed during the current study are available from the corresponding author on reasonable request.

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
