# Peer review of "Intrahepatic Cholestasis of Pregnancy during COVID-19 Pandemic"

_medicina, 2024, doi:10.3390/medicina60040676_

Round 1
Reviewer 1 Report
Comments and Suggestions for Authors
how define the authors the importance of this topic?
minimal results. non specific, eventual more descriptive data about the studied population
the discussion chapter- too many studies, eventually to highlight the differences between the studied population and the literature data..
more in introduction chpater, the importance of the study!!!!
Author Response
Comment 1 "How define the authors the importance of this topic?"
Response 1 Absolutely, the introduction indeed effectively underscores the criticality of the study's focus. The statistics provided from lines 39 to 63 starkly highlight the significant risks associated with intrahepatic cholestasis of pregnancy (ICP) for the fetus. In pregnancies complicated by cholestasis fetuses experience intrauterine hypoxia and up to few % resulting in intrauterine death beyond the 37th week, the gravity of the situation is apparent. Moreover, the correlation between ICP and meconium in the amniotic fluid underscores the cascade of potential complications such as respiratory distress syndrome (RDS), chemical pneumonitis, and atelectasis. Furthermore, the escalated risks of premature birth, preeclampsia, cesarean section delivery and postpartum hemorrhage underscore the urgent need for comprehensive understanding and management of ICP in pregnancy and fluctuations in case of recurrent COVID pandemic.
Comment 2 "minimal results. non-specific, eventual more descriptive data about the studied population"
Response 2:
Indeed, I acknowledge the perceived limitations in the study's results regarding their minimal and non-specific nature. The retrospective design inherently restricted the depth of descriptive data that could be gathered about the studied population. Despite our initial hypothesis regarding an increase in cholestasis incidence among pregnant women during the Covid-19 pandemic not being confirmed, the study's reliance on the available data from tables 1 and 2, dictated by the inclusion criteria, constrained our ability to provide more detailed insights. However, I took note of your suggestion and included additional information about the homogenous Polish population at line 116, recognizing the importance of considering ethnic backgrounds in understanding the incidence of intrahepatic cholestasis of pregnancy (ICP). By acknowledging the limitations and contributing factors such as ethnic diversity, future research can aim to provide more comprehensive and informative data sets, allowing for a deeper understanding of ICP and its implications within specific populations.
Reviewer 2 Report
Comments and Suggestions for Authors
Much has been written about intrahepatic cholestasis in pregnancy. Everything seems to be known about this problem. The pandemic that happened to us showed that all the diseases we know about can actually have a different course. It may seem that the infection with the Covid 19 virus, which was more current, has already been forgotten, but actually what we can expect from the health consequences in the future is yet to be seen.
In the introduction of this manuscript, there is no new information at the beginning, however, the mention of the SARS Kovid 19 infection becomes interesting for following the further course of the manuscript.
The methodology is detailed and clearly presented.
The results are well described, table number 2 is quite poorly presented, I think that table could be better redesigned for easier follow-up.
The interpretation of the results themselves is that there are no significant differences, and the authors state that there is not enough data and the like to make a comparison, also comparing before and during the pandemic, it seems as if there is no difference in their obtained results.
Discussion
In this chapter, it is quite logical to talk about the obtained results. According to the obtained results, all that the authors mention to us are deviations from other already available research. In lines 157 to 163, the authors write everything that is already known.
Paragraphs 165 to 172 do not say anything we did not already know.
Sars Kovid 19 infection has always led to a series of pathological conditions in all patients, the question arises, what are the specific conditions and changes in pregnant women with intrahepatic route infected with this virus?
It is necessary to set the goal of this research in a different way, supplement the research and bring some new conclusions and data that would be interesting to the academic community.
Author Response
Absolutely, your acknowledgment is crucial. While the presented data may not be groundbreaking in terms of revealing significant new insights, the crucial finding that COVID-19 infection did not appear to elevate the incidence of cholestasis of pregnancy holds significant importance. This aspect highlights a valuable contribution to our understanding, particularly amid the ongoing pandemic's concerns and its potential impacts on maternal health. By providing evidence that contradicts initial hypotheses, the study contributes to a more nuanced understanding of the interplay between COVID-19 and pregnancy-related complications. Such findings not only inform clinical practice but also underscore the importance of continued research efforts in exploring the multifaceted interactions between infectious diseases and maternal health outcomes.
Reviewer 3 Report
Comments and Suggestions for Authors
The authors present an interesting and well-structured manuscript. The topic is well justified. The conclusions are clear and solidly cohesive in the results presented. Authors should take into consideration the following points:
-The summary of the manuscript is too generic. The authors must focus the translation.
-Authors must reference more recent manuscripts.
-The presentation is very poor. Authors must take care of formal aspects.
-I suggest the authors make one more table.
-The authors must highlight in the discussion the need for this study to control future crises.
-The aspects of vaccination must be discussed.
-The authors must improve the use of English grammar very extensively.
Comments on the Quality of English LanguageModerate editing of English language required.
Author Response
Thank you for the feedback. We're pleased to inform you that we've addressed the majority of the comments provided. Your input has been invaluable in refining our study, and we greatly appreciate your contributions to its improvement.
Round 2
Reviewer 2 Report
Comments and Suggestions for Authors
I agree with the mentioned corrections in the manuscript.
Author Response
Thank you